# A Tool to Assist in the Analysis of Gaze Patterns in Upper Limb Prosthetic Use

**Peter Kyberd** [1,*,†] , **Alexandru Florin Popa** [2,†] **and Théo Cojean** [3]

1   School of Engineering, College of Science and Engineering, University of Derby, Derby DE22 3AW, UK
2   Institute of Biomedical Engineering, University of New Brunswick, Fredericton, NB E3B5A3, Canada;
    florin.alexandru.popa@gmail.com
3   Université de Lyon, Université Gustave Eiffel, Université Claude Bernard Lyon 1, F-69622 Lyon, France;
    theo.cojean@live.fr
*   Correspondence: p.kyberd@derby.ac.uk
†   These authors contributed equally to this work.

**Abstract:** Gaze-tracking, where the point of regard of a subject is mapped onto the image of the scene the subject sees, can be employed to study the visual attention of the users of prosthetic hands. It can show whether the user pays greater attention to the actions of their prosthetic hand as they use it to perform manipulation tasks, compared with the general population. Conventional analysis of the video data requires a human operator to identify the key areas of interest in every frame of the video data. Computer vision techniques can assist with this process, but fully automatic systems require large training sets. Prosthetic investigations tend to be limited in numbers. However, if the assessment task is well-controlled, it is possible to make a much simpler system that uses the initial input from an operator to identify the areas of interest and then the computer tracks the objects throughout the task. The tool described here employs colour separation and edge detection on images of the visual field to identify the objects to be tracked. To simplify the computer's task further, this test uses the Southampton Hand Assessment Procedure (SHAP) to define the activity spatially and temporarily, reducing the search space for the computer. The work reported here concerns the development of a software tool capable of identifying and tracking the points of regard and areas of interest throughout an activity with minimum human operator input. Gaze was successfully tracked for fourteen unimpaired subjects and was compared with the gaze of four users of myoelectric hands. The SHAP cutting task is described and the differences in attention observed with a greater number of shorter fixations by the prosthesis users compared to unimpaired subjects. There was less looking ahead to the next phase of the task by the prosthesis users.

**Keywords:** upper limb prosthesis; gaze fixation; automatic detection; point of regard; SHAP

## 1. Introduction

The operation of an upper limb prosthesis requires the user to control their device using limited feedback paths [1]. Skill in control is attained as the result of sustained practice and concentration on the part of the user. One of the primary feedback paths used is vision. The hands remain in the view of the operator most of the time and the high quality information supplied about a hand's position and grasp is crucial to successful operation. In contrast, natural hand control is achieved using many different modalities [2] and the hand is generally in the person's peripheral vision only [3].

Gaze tracking is an established technology—in its portable form it uses cameras mounted on the head to record eye movement and the visual field. A computer maps the focus of the gaze onto the scene in front of the subject. The output of the system is a video of the visual field with the foveal point superimposed upon it. This provides information about the direction of the gaze. The visual attention can be inferred from this by observing where the subject looks and for how long. The technology has allowed exploration of

understanding of the role of gaze in reaching, manipulation [3,4], sports [5] and skills acquisition [4]. Other eye-tracking systems can be used to control computer cursors but, as they rest on the desk, they have less use in applications where the subject needs to move about [6]. More recently, this exploration has moved into the investigation of prosthetic grasping [7,8]. For natural grasping, the gaze tends to anticipate the hand, looking to where the hand will be, moving on before the hand has grasped the object to where the hand will go next. In prosthetic manipulation, it has been shown that users tend to follow the hand closely, not looking ahead to the next target until the current one is acquired [7,8].

Employing a gaze tracker is the first step towards understanding how prostheses users look at a task involving their prostheses. Potentially, this will allow improvement in the design of the control of prosthetic hands while reducing the cognitive burden. Analysis of the recorded scene requires human intervention. Generally, it is a human operator who goes through the video data frame-by-frame. They must identify and mark the key items and events in the visual field. This is clearly time-consuming and dependent on the skills of the operator. Computer vision techniques are available to interpret the scene, but machine learning needs to be able to segment the data into significant actions and to aggregate real, noisy data into statistically similar groups. Thus, it needs copious amounts of data to learn from [9]. In contrast, similar to the development of low-cost, motion-tracking systems (such as the Kinect for home gaming), it is possible to create a simplified system that will speed up the image analysis without the expense of highly sophisticated computers and software [10,11].

This paper describes a system designed to speed up the process of processing eye-tracking data for the analysis of prosthetic hand function. The program uses a standardised set up, including analysis of activity of daily living tasks. A computer is able to identify the key areas of interest and track them in the visual field throughout the activity with minimal human intervention. It is believed that this is the first time this assisted analysis has been created for this application.

### 1.1. Background

The study of gaze behaviour and visual attention for upper-limb activities of daily living (ADLs) has been conducted since the technology became compact enough to make field studies practical [12]. More recently, it has garnered the attention of the prosthetics field, with an initiative from three groups [7,8,13]. This was later expanded to the wider community [14–16]. All of these studies have required hand-labelling of the video data by a skilled operator. The overall goal of this program of study was to assess the visual attention of a prosthesis user when faced with a new task, and to gauge the influence of the usability of upper limb prostheses by observing the eye gaze patterns of users of prosthetic hands. The aim of the project detailed herein represented a first step—to develop a method for assessing visual attention with minimal human intervention.

In commercial systems, the video data from cameras mounted on the head are combined by a computer to map the point of regard (PoR) onto the image field. One camera records the motion of the eye and uses the position of the pupil to derive the PoR. This is mapped onto the image taken by a second head-mounted camera of the scene. Before use, the system is calibrated.

Generally, to analyse the videos generated, the experimenter has to step through the video frame-by-frame, identifying items of interest and inputting them into a computer for numerical analysis. This is laborious and time-consuming; it relies on human interpretation of the areas of interest (AoIs) in the scene and is dependent on the level of skill and attention of the observers [17]. Computer systems and image-processing software are now able to analyse complex video images through forms of machine learning [18,19]. However, to achieve an adequate level of proficiency to be useful [9], the systems require many examples of the test data to train the system. Studies with prosthesis users rarely generate enough data to train an advanced machine learning model sufficiently. The number of repetitions each subject would have to perform is unrealistic. This project created a system that did

not require the computer system to be that competent or to require many examples to train the software. Instead, the software was designed to extract key items of interest from the video of the visual field, with the operator marking the first instance of an AoI and then the computer being able to track it in subsequent frames. The automatic identification was based on three details: the colour of the object, its shape, and that an object cannot have moved far between successive frames. This approach has not been used previously for this sort of application.

The project was not focused on generating new ideas and technology in the computer vision field, but to use existing tools to extract information. Thus, an interface was created that used tools within Matlab (Mathworks, Natick, MA, USA) to extract sufficient information to assist in the identification of a range of points of interest in the visual field. Data-processing and analysis used a combination of pre-existing Matlab functions and custom-written routines.

### 1.2. Assessment Framework

Assessment of complex activities is a compromise between extreme abstraction (which is easy to measure but may not provide realistic results) and verisimilitude (which is highly variable and makes it hard to draw general conclusions about a population) [20]. For this application, an assessment form was chosen that controlled the task and the environment to assist the computer in tracking the task while ensuring it had clinical validity.

While some experiments have employed simulated prosthesis users as subjects [7,14,21] and abstract tasks, for this study an assessment tool was chosen to allow for the measurements to be made while the routine users of prosthetic hands performed simulated activities of daily living (ADLs). The Southampton Hand Assessment Procedure (SHAP) was chosen because it has qualities that make it particularly appropriate for this application [22]. SHAP is a tool validated for the assessment of users of prosthetic hands. It measures the functional capabilities of the user and hand *in combination* [23]. All functional tests are a compromise between the practical (what can be achieved in the time, space and budget) and the accurate use of real-world operations. SHAP uses a series of standardised tests based on tasks that were already proven to be repeatable and validated [24]. These tasks are timed by the operator as this too increases the reliability of the measurement [24]. A score is produced which is based on the Malhanobis distance of the times of the subject from a standardised set of measurements. The overall score is out of 100, with scores above 95 being the range for the general population. There are eight abstract tasks (picking and placing abstract shapes) and fourteen simulated ADLs. The layout of the tasks is controlled by a form board. SHAP has been used in a wide range of conditions, including prosthetic limbs [14,25,26], and in grasp kinematics [27,28] (in conjunction with motion analysis systems).

A SHAP task is set up on the form board. The start and end positions for all elements are marked on the board; the board also fixes the timer in the centre. Participants are instructed to perform the tasks with the hand under test (the non-dominant or the prosthesis in these experiments). The subject presses the button on the timer, performs the test and then presses the button again to turn off the timer and complete the task (Figure 1). This time is entered in a database which calculates the overall score and the score for each grip form [22].

For this application, SHAP has the additional advantage of providing a framework that allows the computer system to make assumptions about the underlying structure of the tests. The form board holds the parts of the task in a fixed starting point. Each surface of the form board is a strong standardised colour (red for ADLs and blue for abstract objects). The timer button is in the centre and is blue surrounded by the grey of its case. This makes the prediction of where points of interest lie in the first frame and their extraction from a noisy video field easier. Thus, any computer-based search can start with these features assumed, e.g., colour, contrast and relative position in the visual field, and search for known shapes and colours in the most likely places.

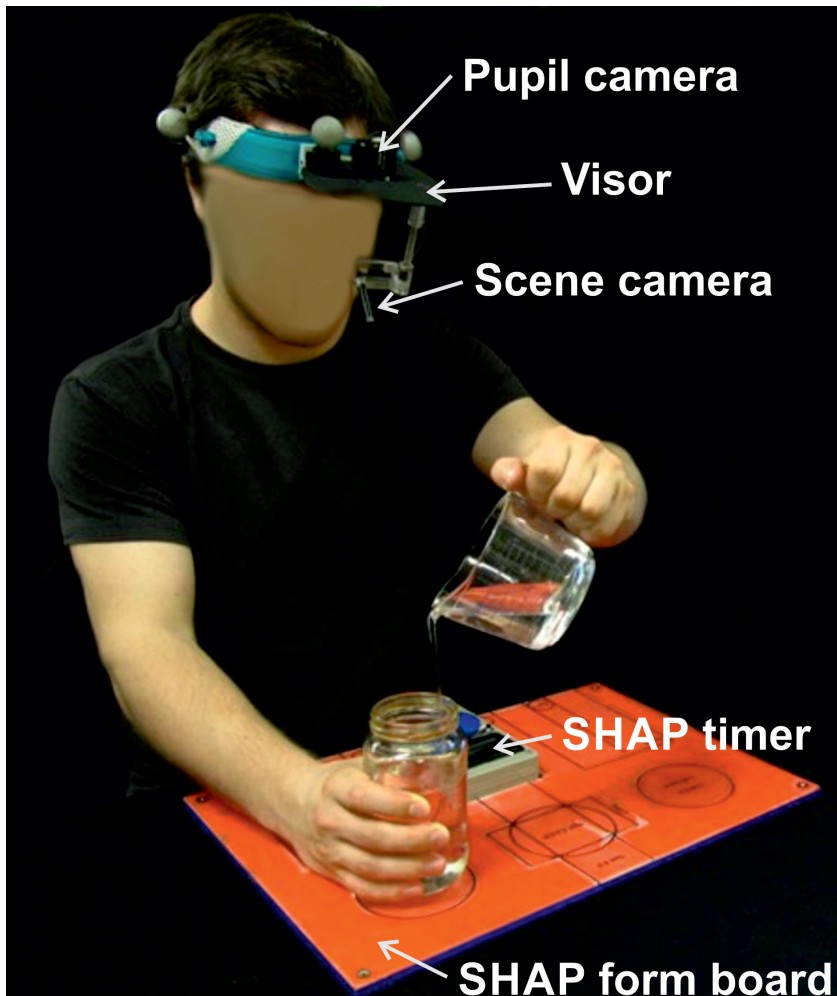

**Figure 1.** Experimental set up: Shown is the ISCAN visor that holds the pupil and scene cameras. The pupil camera reads the position of the pupil via a mirror; these data are used to reconstruct the PoR in the scene recorded by the second camera. The SHAP task shown here is the pouring task with the form board and the timer in front of the subject.

*1.3. Task*

For this paper, the SHAP task chosen for consideration is the cutting task. The task is laid out with a cylinder of plasticine placed in the middle of the form board beyond the timer. The knife is placed on the side of the hand under test. A test begins with the user starting the timer with the hand under test. The subject is allowed to pick it up with their contralateral hand and place it in the prosthesis (if they are prosthesis users). They then grasp the plasticine with their contralateral hand and cut the strip before replacing the knife beside the timer and turning the timer off with the test hand.

The gaze information was analysed in relation to a series of areas of interest (AoIs). Every activity has different AoIs, depending on the form of the ADL that is captured in the scene. For example, for the cutting task, the points of regard (PoIs) are the knife, the plasticine, and the timer button. Investigation of the visual attention was then achieved by evaluating the amount of time the PoR was fixated on the specific AoIs throughout a given activity.

## 2. Method

Subjects were asked to perform simulated ADLs using SHAP. During the tasks, the visual attention was recorded using a head-mounted eye-tracker, ISCAN (ISCAN, Inc., 21 Cabot Road, Woburn, MA, USA). The ISCAN system collects video data of the scene,

together with the coordinates of the pupil of one eye. The ISCAN identifies and suppresses saccades (quick movements of the eye to a new PoR) and maps the PoRs onto the video of the image field. This is stored in a standard video format. It is these data of the visual field and the point of regard that were used as the input to the analysis software described here.

A schematic of the entire system is shown in Figure 2. The guiding principle of the software design was that it should be simple to use and not need extensive development of specialist machine learning software. The initial identification of different areas would be made by the operator on the first image of the sequence and the computer would be able to extract sufficient information about the image to track through the rest of the video. Should the computer lose track of an area of interest, the operator could help the program re-acquire the area through simple interactions with the computer.

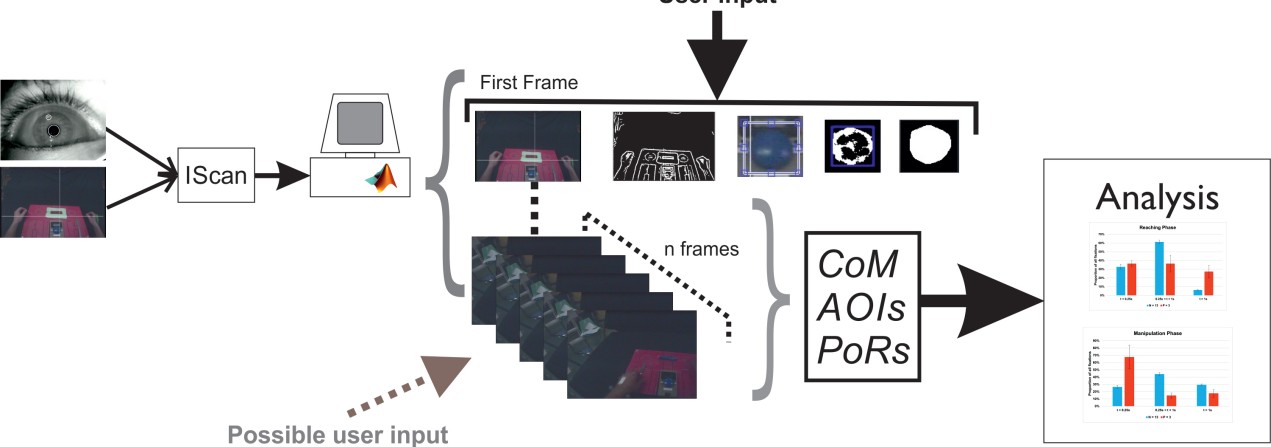

**Figure 2.** Schematic of the process. The ISCAN system uses two cameras to provide point-of-view data. This video is then analysed for information on the point of view and areas of interest in each frame; having been trained on the areas of interest in the first frame, the system steps through frame-by-frame. The data produced are then analysed for insight into gaze behaviour.

The custom software presented the first frame to the operator who used a mouse to identify the key areas of interest in the particular task. The software then identified the colour and shape of the object and subsequently stepped through the video frame-by-frame using these data on the colour and shape to identify the AoIs in each new frame. As the AoIs cannot move very far between successive frames the computer need only search in a small area close to the position of the object in the succeeding frame. This reduces both the search time and the chances of another item being misinterpreted as one of the PoIs. If the computer had not found the object within the search area, it then requested the operator to point to its location. Once the AoI was found, then its centre of mass (CoM) was recorded and the computer went on to identify all other AoIs in this frame. Once complete, the computer moved on to the next frame, and continued frame-by-frame until the end of the video was reached.

The ISCAN was worn on the head and the cables were run to a computer. The video stream information was captured at a frame rate of 29.97 frames per second. The area surrounding the table was curtained off to remove unwanted additional images in the peripheral vision. The view generated by the system is shown in Figure 3. This image is of the form board and the timer. The cross-hairs for the point of regard are visible. This will be referred to as the 'primary image' on which the analysis software bases its initial decisions; the results will be shown throughout the rest of the paper.

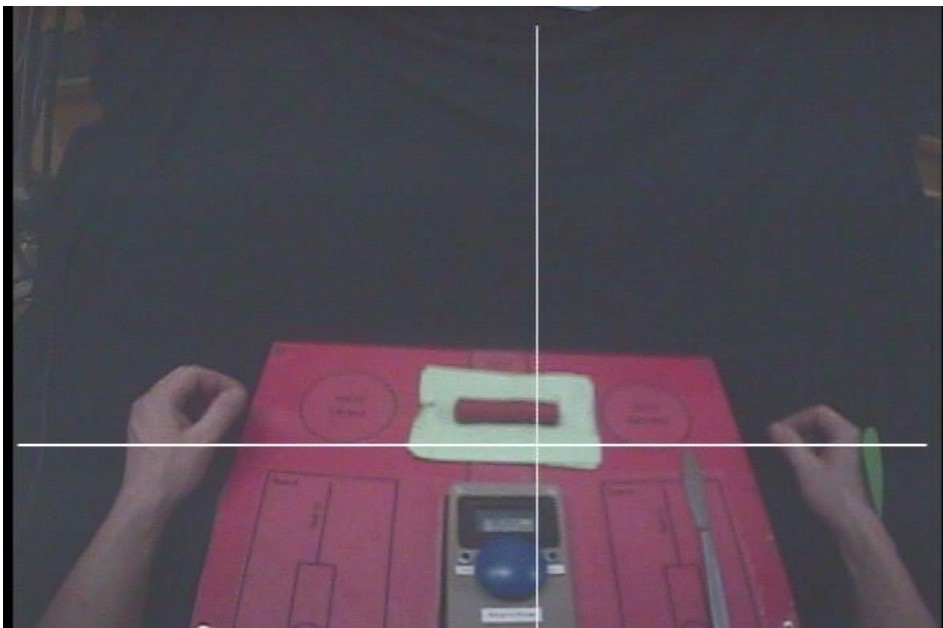

**Figure 3.** An example of the image created by the ISCAN system. The form board, subject's hands, and the cross show where the subjects gaze is focused. The form board is set up for the cutting task. The object to be cut is in the top centre, the knife is to the right, and at centre bottom is the timer with the on/off switch in blue.

### 2.1. Image Processing

The video is a sequence of images or frames. The analysis was undertaken frame-by-frame, starting with the first image. The system identified and tracked the desired objects within each image. The code was not constructed to work without *some* human intervention. The software coordinated the identification of the key points of interest in the first frame by the operator and the computer tracked the movement through the visual field in subsequent frames using distinctive visual features. The software extracted the boundaries and coordinates of the AoIs from the videos captured. It used established image-processing techniques [29].

The task was divided into three parts:

1. Train the system on the first frame
2. Iterate through subsequent frames using the information to locate the PoIs and record their coordinates
3. Calculate the relative position of the gaze from the PoIs and determine a glance or fixation and display the information graphically

Details of the first two stages are reported here and some initial results from the analys are shown.

Training on first frame

The system presents the first frame and prompts the operator to identify all the AoIs for the particular task. The system then extracts the features, including colour, shape and centre of mass, and stores them, as shown in Figure 4a.

Identification on subsequent frames

For all frames in the video, the basic sequence remains similar until the end of the video: Extract the cross-hair position using the feature information and search windows to detect all AoIs. If one cannot be found, prompt the user for input, otherwise update the features and create a new search window (Figure 4b).

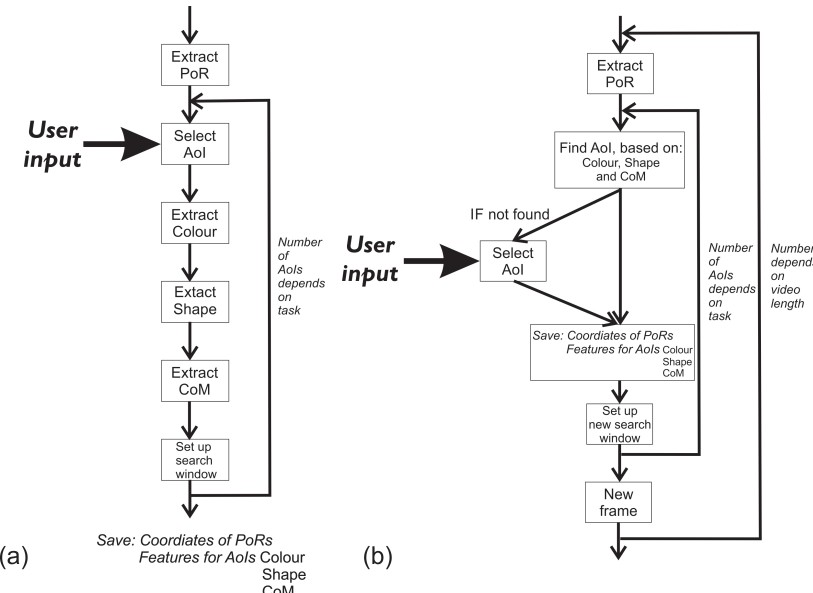

**Figure 4.** Extraction of AoIs and PoRs. (**a**) Training the system on the first frame for the features, colour, shape and movement. (**b**) Extraction of AoIs in subsequent frames.

2.1.1. Object Acquisition

The first step to acquire the data is to locate the objects in the scene. This is performed by identifying their boundaries and, hence, recognizing the items as separate entities. For this, two spatio-temporal methods were employed [30], one based on colour and the other on edge segmentation.

*Colour-based segmentation* This creates segments on the basis of similarities in colour intensity. The intensity is extracted from maps of the fundamental colours (red, green and blue) [31]. The precise value of the intensity depends on the particular illumination level and a single object can possess different colour attributes at different times. To circumvent this, a measure of 'nuance' was implemented. The colour descriptors were derived from the normalized RGB channels, and were referred to as 'redness', 'blueness' and 'greenness'. The code accepts a range of colour values around the core values of red, green and blue. For an 8-bit map the red colour is 255 and the other two colours are zero Figure 5. However, to have 'redness', the red colour needs only 70% of the maximum value and a maximum of 20% of either or both of blue and green. To ensure that there was a full range of colours to analyse, the ranges were first normalised. The mean intensity and standard deviations for the RGB were calculated and the nuanced images produced. Two nuanced colour maps are seen in Figure 6, together with the original image.

```
FOR Every pixel in the frame
IF (Red value IS GREATER THAN 255 x 0.7)
THEN
IS RED
ELSEIF (Blue value IS GREATER THAN 255 x 0.7)
THEN
IS BLUE
ELSEIF (Green value IS GREATER THAN 255 x 0.7)
THEN
IS GREEN
ELSE
No action
ENDIF
ENDFOR
```

**Figure 5.** Colour map decision algorithm.

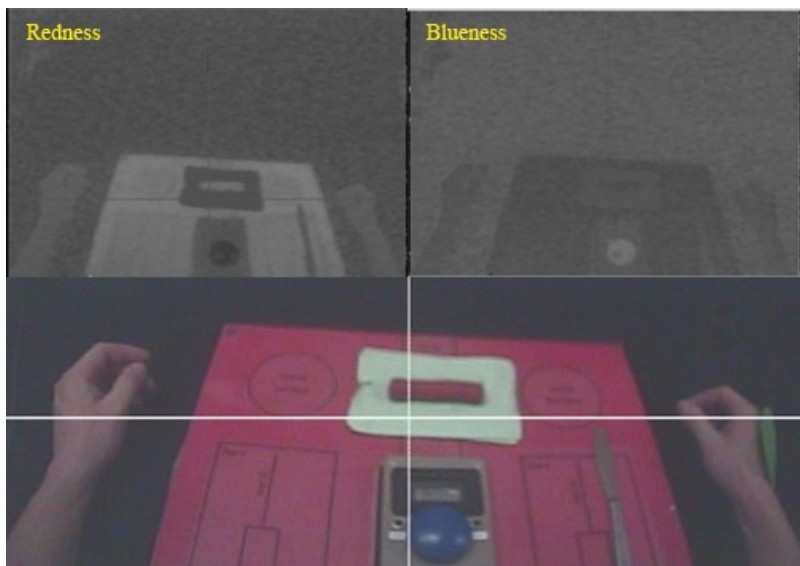

**Figure 6.** Red and blue 'nuance' colour images, compared with the primary image. Colour of interest is shown as white in the nuance colour images.

*Edge detection* As colour segmentation cannot discriminate between similarly coloured objects, the next stage was to use edge-detection techniques. Edge detection is based on detecting sudden changes in intensity [29]. Thus, the derivative of the intensity in the image locates the majority of the edges. The interface applied a Canny edge-detection method from the Matlab signal-processing toolbox. This involved application of a Gaussian filter on the data and then calculation of the derivative of the image [32]. Figure 7 shows the edges on the primary image.

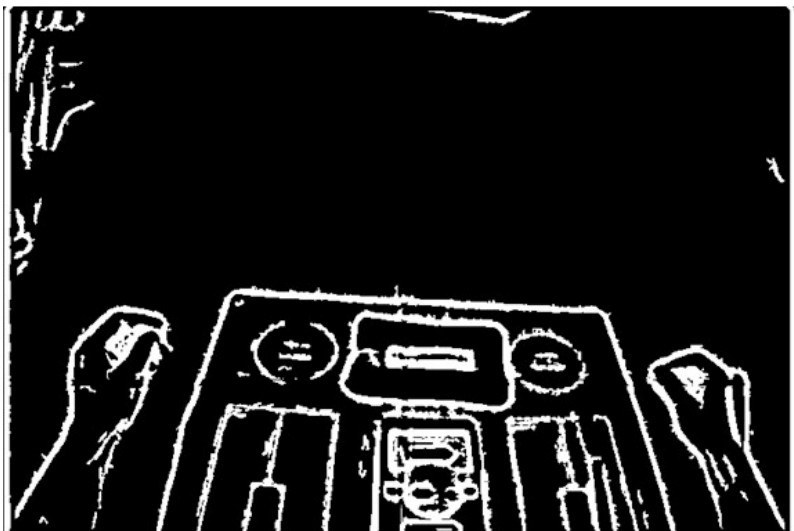

**Figure 7.** Result of applying edge detection on the primary image.

An example of shape extraction for the button is shown in Figure 8. The user inputs a selection of the button by pointing the mouse at any part of the button. The computer constructs a bounding box around the area that is roughly the same colour. At the same time, the convex shape based on the edge-detection algorithm determines the size of the bounding box. The colour mask is then applied to the shape. The top of the button has a flare from the lights of the room so that to the camera it is not all the same colour, but, by combining the colour characteristics and the the edge detection results, the final shape approximation is created. In subsequent frames, the bounding box can be applied and the button reacquired.

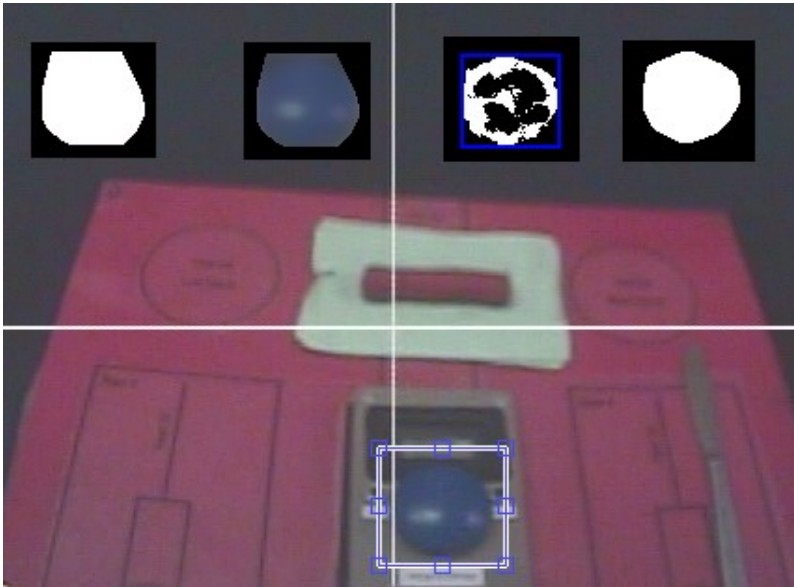

**Figure 8.** The process of assisted segmentation. The user indicates an item of interest (the time button), and, based on the edge detection and colour matching results, the computer can track the position of the button in the frame.

*Point of Regard Tracking* Once the bounding box is defined around the object, it is possible to define when the subject is looking at that particular area of interest. This occurs when the cross-hairs are inside or close to the bounding box. From this, the gaze behaviour can be built up from analysis of the individual frames. 'Close' to the bounding box is defined when the co-ordinates of the PoR are within 10% of the size of the object. In this way, the size of the target object is taken into account.

### 2.1.2. Search Window

An object is tracked by being identified in every new frame. The search time to identify an object can be reduced by restricting the search area to that in which an object can be reasonably be expected to move in the time between two consecutive frames. For each new frame, the classifier aimed to identify the objects based on a maximum reasonable distance between the observed feature vectors.

The sequential frames are 33 ms apart. An object is placed in front of the subject, between them and the form board, up to 300 mm from the camera. The form board is 430 mm by 310 mm. A maximum velocity was set to be 0.2 ms$^{-1}$ (four times the peak reaching speed recorded by Bouwsema [33]). If it is moving perpendicular to the visual field, this corresponds to 33 mm between frames. The furthest edge of the form board occupies half of the frame, and, for the output of the ISCAN, this corresponds to 720 by 540 pixels (see Figure 3). Thus, the range limit is one tenth of the width of the visual field (72 pixels). The search window for subsequent frames is set to be larger by 72 pixels all round. If the object is not found within this window, the interface is referred back to the operator for a new entry.

### 2.1.3. Data Extracted

For analysis of the action, the characteristics that were then extracted from the video data were the Euclidean distances between the PoR and the CoM and bounding boxes of the AoIs. The duration of the individual fixations was measured by counting the number of frames the PoR was within an AoI. These data are sufficient for concluding whether a subject is fixating an object or is glancing at it. Based on previous work [34,35], a fixation was defined as a time on the target greater than 200 ms (seven frames). Once the data are obtained, it is possible to gain insights into behaviour.

2.1.4. Subjects

Fourteen subjects without limb differences were recruited from the staff and students of the University of New Brunswick (UNB), including nine males and five females. Nine were right-handed (mean age 29, range 18 to 48), and four users of prosthetic limbs volunteered through the limb clinic at UNB (ages not available). Research ethics approval for the experiment number was obtained from the UNBs Research Ethics Board (REB2010-099). All users had absences below the elbow and used standard single-axis myoelectric limbs with friction wrists and were allowed to adjust the orientation of their prosthesis before each activity. A single prosthesis user performed the test twice, first with their prosthesis, and, on the second time, with their contralateral hand, allowing additional comparisons.

After the ethics processes were completed, each subject donned the ISCAN headset. Prior to the tests, the system was calibrated. Subjects were then asked to perform the six SHAP tasks (pouring water, coins, cutting, abstract lateral, page turning, operating a zip) (for details, please see [22,23]). Subjects then removed the eye-tracker headset and performed a SHAP test according to the standard protocol.

## 3. Results

For this paper, the emphasis is on the cutting task. The results of the overall SHAP tasks and other details are included in a separate publication [36]. Due to limitations in the system, gaze data were not available for some instances (see Section 4). In a test of 100 gaze video sequences, the semi-automated tracker only lost the various targets when the ISCAN device lost track of the gaze. There were 18 users with 6 tasks, and 8 sequences where the fixation data were missing.

### 3.1. General Observations of the Task

Non-impaired subjects tended to look at the button as they were reaching for it, but quickly changed their focus onto the knife before their hand had reached the button. Similarly, their gaze moved ahead to the plasticine while picking up the knife. Subjects may have looked briefly at the blade to ensure it was in the right position, especially if they had not handled the knife before. In contrast, the impression from observing the video was that prosthesis users looked at the button until it was pressed and attended to the knife until they had it firmly in their grasp. They then fixated on the plasticine until they had finished cutting it, and so on.

### 3.2. Video Information

To assist in the understanding of the usual behaviour afforded by this tool, data for a single unimpaired subject were analysed. The processed data were represented in the 2D-plane of the image, allowing different ways to represent the data—examples are given here. Instead of showing the data relative to the static field of the observer once acquired, it is possible to transform these data relative to different points of regard. The software can assess the relative position of the point of regard (PoR) with respect to the centre of mass (CoM) of each object of interest (OI) in the scene. Figure 9 shows an example of the position of a single OI (the blue timer button, positioned here top left). For each frame, the COM is plotted at the origin ('stabilized' CoM). The PoRs are show as adjusted relative to the button. Their offset to the origin, relative to the CoM, is plotted on the vertical and horizontal axes. Each cross represents the PoR in a specific frame in the video. The blue rectangles surrounding the origin represent the bounding boxes of the button, i.e., the minimal area of rectangles that completely confine the OI. There are multiple bounding boxes for each frame; hence, a heavy blue line. The crosses are the PoR and they are shown in blue outside the bounding box and red inside, suggesting that, for those frames, the subject was fixating on that OI. Using this technique, one can observe that the subject does not attend to the button except when they are reaching for it.

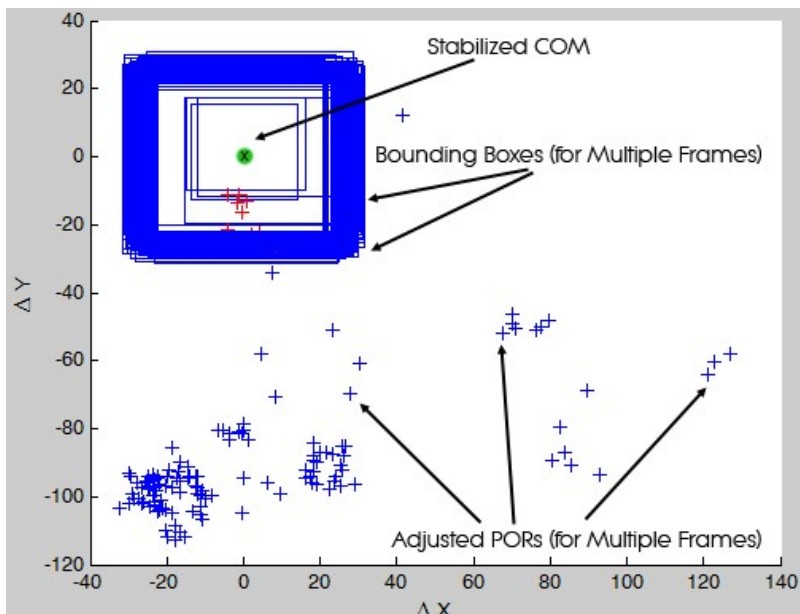

**Figure 9.** Relative PoRs to CoM coordinates for the cutting task for an unimpaired subject. The timer button is stabilised as the centre of mass and is at the origin. Note: Upper left. The multiple PoRs are then shown relative to it. The red crosses are when the attention is on the button.

Figure 10 shows the time series of the same sequence. The horizontal axis is the frame number; hence, the time. The vertical axis is the distance from the blue button CoMs. Red circles mark the frames when the PoR was inside the bounding box of the reference object (fixating on the button). This is an analysis of 150 frames or approximately five seconds of recorded video. The subject looks at the button only once. Figure 11 shows the Euclidean distances between the PoR and the CoM of another tracked object within the scene (the knife). It can be seen that the PoR is never within the bounding box of the OI, so the subject never looks directly at the knife, but can see it in their peripheral vision only.

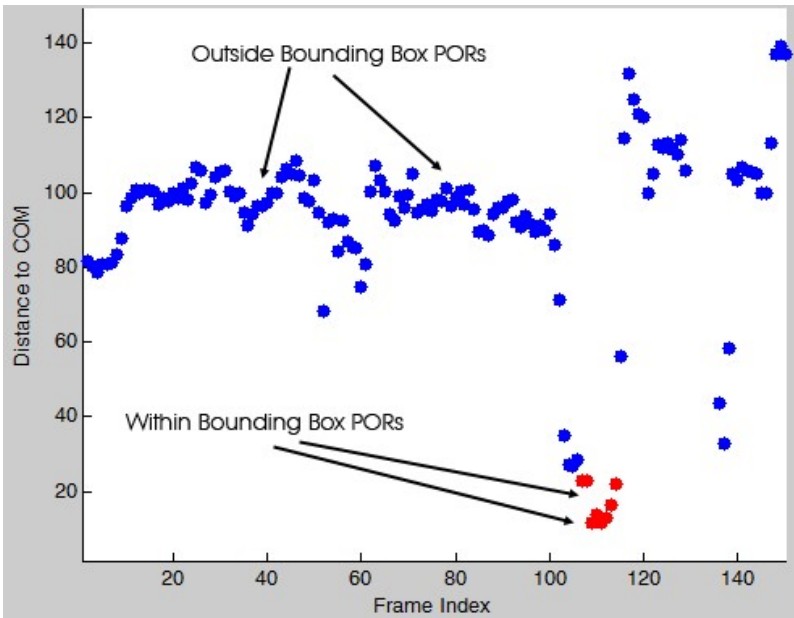

**Figure 10.** The motion of the point of regard (PoR) relative to the blue button (CoM) for an unimpaired subject. The Euclidean distance is on the y-axis and the frame number (hence time) is on the x-axis. This unimpaired subject looks at the target before they reach towards it.

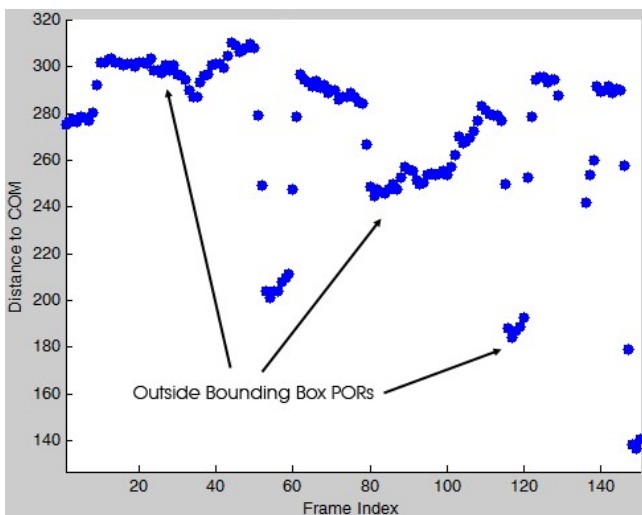

**Figure 11.** The motion of the point of regard (PoR) relative to the knife (CoM) for an unimpaired subject. The Euclidean distance is on the y-axis and the frame number (hence, time) is on the x-axis. The unimpaired subject never needs to look directly at the target, merely keeping it in their peripheral vision.

### 3.3. Analysis of the Statistical Data

To demonstrate the other results that can be obtained from this analysis an aggregation of the data for the cutting task is shown in Figure 12. It depicts the proportions of the average number of gazes for both groups. The data are divided into the reaching and manipulation phases. The data are further divided into three temporal phases: below a quarter of a second, from that to one second, and gazes longer than a second ('glances', 'looks', and 'stares').

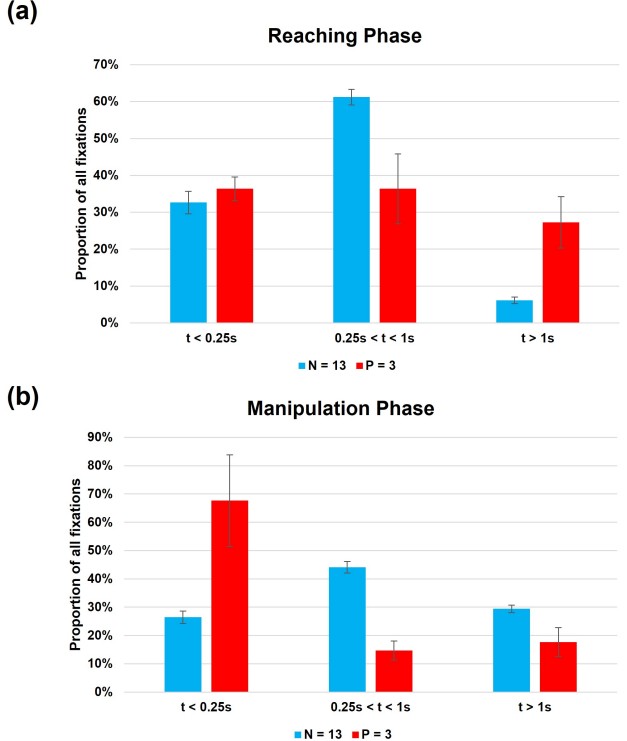

**Figure 12.** Number of fixations for the cutting task divided into three phases: Below 0.25 s, between 0.25 and 1 s, and greater than one second. (**a**) Data for reaching phase, and (**b**) the manipulation phase. Error bars are standard deviation across subjects.

The results show that, when they employed their prosthesis, the users showed a greater range of glances across the visual field. The proportion of the number of 'glances' was similar in both groups, but the users fixated on the target for longer. Of note is that the users took longer overall to perform the tasks. The unimpaired average trial duration was $4.8 \pm 1.5$ s, with $6 \pm 3$ fixations, while, for the prosthesis users, it was $20 \pm 12$ s with $14 \pm 10$ fixations. During the manipulation phase, the number of glances was far higher in the prosthesis population, while the unimpaired subjects stared directly at the plasticine.

## 4. Discussion

The system proved to be sufficiently reliable for much of the process to be automated. There was a greater number of times when the ISCAN system was unable to track the gaze of the subjects than when the custom software could track the OIs. This limitation can be reduced by ensuring the ISCAN system is set up correctly at the beginning. There are numerous factors that dictate the success of the system, including the position of the tracker on the head of the subject, the level of illumination, the shape of the subjects' orbits, and whether the visor carrying the scanner had been inadvertently moved by the subject. The experimenter needs to be vigilant in detecting and reducing these problems.

### 4.1. Gaze Tracking

This simplified approach to the automation of the analysis of the visual field is novel in this application to prosthetics gaze tracking. The time to undertake the basic analysis was much reduced and the data produced became available for many different forms of subsequent analysis. It has been shown that is possible to evaluate gaze behaviour by identifying the frames at which the PoR is fixated at specific OIs or AoIs with less user input and reduced requirement for reliability of the skills of the operator compared with a standard process [17].

Different insights can be gained from the different ways of creating and displaying the data. Time-series data can allow reading of a single activity, and the collection of numerical data allows for aggregation. By counting the number of consecutive frames, the duration of each fixation can be extracted, and a measure of the amount of attention paid to any particular point of regard derived. An example is shown in Figure 10. The PoR was fixated at the timer button for ten consecutive frames. This amounts to 0.66% of the total time analysed (150 frames or roughly 5 s). Whereas in Figure 11, the local minima (seen in frames 50 to 60, 115 to 120 and around frame 150, for the knife), show that the PoR quickly approached the CoM of that object of interest. This suggests either a glance towards the OI or a focus towards another OI that is in the vicinity. Study of the video frames shows there is nothing of interest at these points; thus, the subject is simply glancing *towards* the target.

Data aggregation allows for a nuanced analysis of the population data. One example from this task is the amount of time the operator looks at the button compared with the time regarding the knife. Comparing Figures 10 and 11 with the number and length of different fixations (Figure 12) shows how much attention the subjects need to attend to the sub-tasks.

The key focus of interest for this investigation is comparison of the actions of prosthesis users compared with the majority of the population. In both groups, reaching represents a similar percentage of glances at objects (times less than 0.25 s). The prosthesis users then 'stare' at the objects (such as the button) for far longer than the unimpaired (percentage of fixations greater than 1s, compared with mid-length 'looks' between 0.25 s and 1 s). During the manipulation phase, this is very different, with the users spending more than 60% of the time glancing between the prosthesis and the task itself, while, once again, the unimpaired subjects spend the majority of the time looking at the plasticine. The inference from this is that the user of a prosthesis needs to monitor the prosthesis and the objects equally, moving their eyes away from the task to the hand and back, checking that the prosthesis is still retaining the object. This results in many more short fixations relative to the unimpaired subjects, who can rely on touch feedback to tell them if the object is slipping from the grasp.

### 4.2. Head Orientation

Humans can acquire visual information using different behaviours. They can look directly at an object (foveate), glance at it, or use peripheral vision to check on the location of important, but secondary, details [5]. Additionally, the individual can divert their gaze towards the object. However, for more concerted focus they can rotate their head towards the object and fix it in the centre of their visual field. This is likely to be associated with longer fixation times or when priority is given to the task. To extract this information from eye tracker data, one solution would be to require the computer to numerically transform the entire visual field and obtain the details of the rotations of what is seen [37,38]. Using the objects within SHAP makes the analysis far simpler. If the timer's bounding box remains in the lower centre of the frame then the subject is pointing ahead. Only if the button moves away from the centre of the frame and the PoR moves towards the centre is the subject concentrating on the activity.

This difference can be observed by plotting all the PoRs on a single two-dimensional field. If the subject turns their head towards a target, the PoRs will be closer to the centre of the image, Figure 13. An illustrative comparison can be made between the behaviours of the single subject that performed the test with their prosthesis and their unimpaired hand. With their contralateral side, the subject kept their head pointing forwards and used peripheral vision to track the activity (green crosses). With their prosthesis, the operator turned their head *towards* the knife on the left side of the form board, putting it closer to the centre of the visual field (red crosses towards the right side). The gaze can be seen to cover more of the visual field as they look about more from the greater spread of red crosses. This, coupled with the difference in fixation durations, paints a picture of someone with a far greater focus on the task.

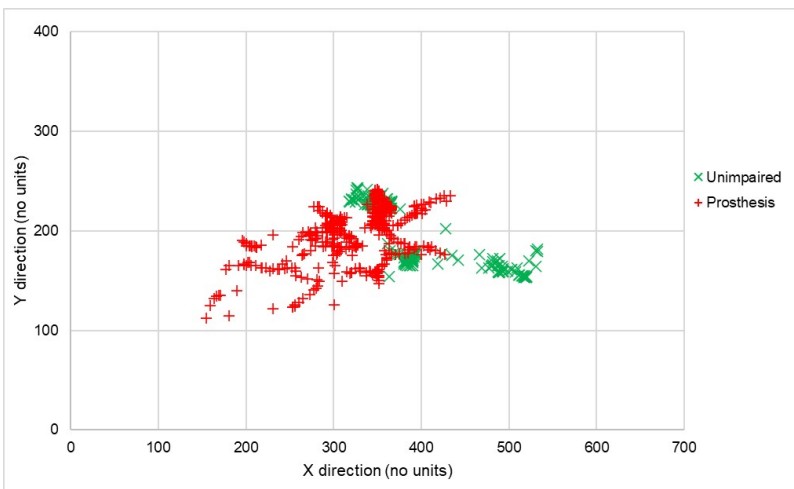

**Figure 13.** The range PoR across the visual field for the unimpaired arm and the prosthesis. Values are dimensionless numbers from the digitisation of the visual image. Green crosses are for an unimpaired hand, red are for a prosthesis.

Finally, the point of view of the subject provides clues as to how they compensate for the lack of flexibility in some of the joints in their prosthesis. Users of prosthetic limbs tend to compensate for the lack of flexibility in the wrist and forearm by using a combination of humeral abduction and trunk sway instead of flexion/extension and pro-supination [39]. This motion tends to cause the visual field to rotate. This is clear in a comparison of frames during the pouring task in Figure 14. The first example is an early frame and the second is taken during the pour, showing considerable rotation of the body to the right in compensation.

The choice of task has a strong influence on the applicability of the assessment. As has been indicated, the greater the abstraction of the task, the more general the conclusions, but these are more removed from real activities in the field [20]. SHAP was designed to create

a balance between the contrasting pressures, using clinically valid tasks that are sufficiently well-controlled to allow broader conclusions to be drawn about manipulation [22]. Other assessment tools have been developed that could also employ similar techniques to automate the process of analysis [40–42]. Using SHAP tasks restricts the activity to small-scale indoor manipulations, but, as the hand is generally used in this circumstance, this is not too great a restriction.

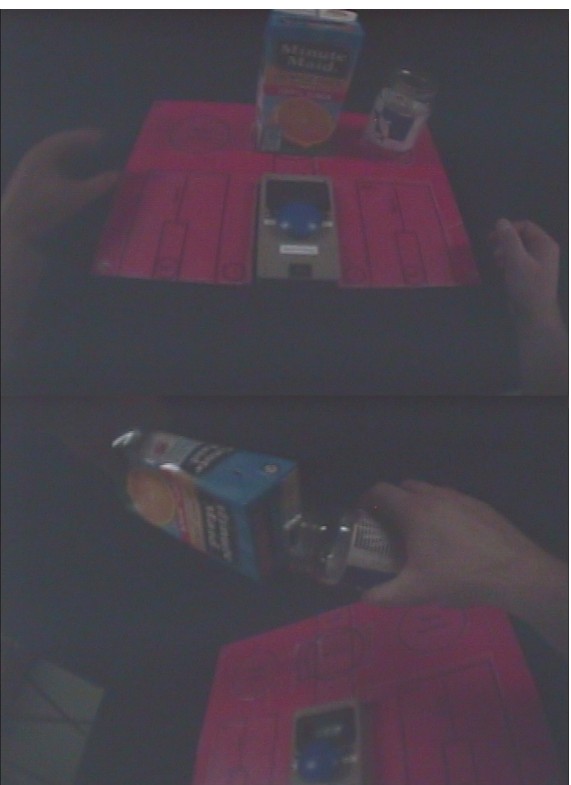

**Figure 14.** Example of the rotation of the visual perspective as a prosthetic user employs a combination of trunk sway and humeral abduction to compensate for a lack of pro-supination in their prosthesis. The subject has lent to the right as they poured the carton. The image rotates to the left. The first image is the initial set up and the second during the pour.

An additional advantage of this method is that it allows a measure of the closeness of the gaze to an object to be quantified. With conventional video analysis, the points of interest are marked by hand. If the size of an object of interest is considered, it has to be arbitrarily assigned by the operator too. This software derives a size from the visible outline of the object and a centre of mass is based on the image itself. Any fixation or a glance in the peripheral vision can be measured and recorded repeatedly and objectively.

*4.3. Refinements*

While the system proved satisfactory, there are a number of possible refinements to automate the system further.

It is possible to detect trunk sway and to measure its timing or extent from the data. Again the choice of a SHAP task makes it far simpler as the displacement of the button from the lower centre of the visual field, combined with the angle of the form board, is sufficient to detect the motion without needing to derive the precise angle of flexion.

The system does not rely on any particular form of visual information, such as how the PoR is defined or shown on the images, which is a particular characteristic of each gaze-tracking system., This tool has been shown to work with other gaze-tracking hardware (e.g., Dikablis Professional Wireless Eye Tracker [43,44]).

A refinement to the code would be to use the context of the known activity (such as cutting) to automatically search for the knife, button and plasticine in the early images, based on their known colour, where they should be in any SHAP sequence, although this level of automation will make the tool less general for use with other tasks. Additionally, it is possible to use the motion data of prior frames to restrict the search box further, predicting position based on the preceding motion of the last few frames, rather than using an arbitrary constant maximum speed. Again this level of sophistication was not found to be needed in this application.

This interface has been used to automate the process of studying gaze during the learning process. When performing an unfamiliar task, subjects tend to fixate for shorter periods of time compared with routine operations; thus, it is more important to have objective measures of gaze as provided by this program.

## 5. Conclusions

Using a standardised format for a task, it is possible to create a simple interface that tracks the points of interest in a video sequence that records the motions and the point of regard within the visual field. Once the system has been given some points of interest in the first frame of the video sequence by the operator, the system can track their motion through the activity of interest, determining how the subject looks at the key points on a visual field during the activity, and indicating that prosthesis users devote more focus and attention to a task. This system has been used to analyse the data from the tasks and to investigate the behaviours when a task is new to the operators.

**Author Contributions:** Conceptualization: P.K. and A.P.; methodology: P.K., A.P. and T.C.; writing—original draft: A.P. and P.K.; writing—review and editing: P.K.; project administration: P.K.; funding acquisition: P.K. All authors have read and agreed to the published version of the manuscript.

**Funding:** This research was funded by the National Science and Engineering Research Council of Canada, Discovery Program.

**Institutional Review Board Statement:** University of New Brunswick Research Ethics Board, REB2010-099.

**Informed Consent Statement:** Informed consent was obtained from all subjects involved in the study.

**Data Availability Statement:** Not applicable.

**Acknowledgments:** The authors would like to thank the volunteers and the staff of the Atlantic Clinic and Ali Hussaini and Philippa Gosine.

**Conflicts of Interest:** The authors declare no conflicts of interest.

## Abbreviations

The following abbreviations are used in this manuscript:

| | |
|---|---|
| ADLs | activities of daily living |
| AoI | areas of interest |
| OI | object of interest |
| PoR | point of regard |
| CoM | centre of mass |
| SHAP | Southampton Hand Assessment Procedure |

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
