# Peer review of "A Tool to Assist in the Analysis of Gaze Patterns in Upper Limb Prosthetic Use"

_prosthesis, doi:10.3390/prosthesis5030063_

Round 1

Reviewer 1 Report

The paper is important, will be more important if you also attach the code, and make it more general not just to prothesis but to any visual attention acuasition

Still, it is not ready for publication - requires major majot revision

Abstract not very clear

Define better the problem: what are the input of the problem?  Video of the face? The scene? The location of the hand? Iassume it is both from an eyetracker – so write it down.

Make the sections as two:

Introduction

Related work – if you want

But introduction need to be part of the introduction

Missing in introduction

Contribution of your work – not clear! You need to refer the contribution to existing similar competing teghcnioglies

Methods

-          please add acheme of the experiment: human + eye tracker + scene

-          Please add flow of the algorithm in a figure

-          No formulas. No clear algorithm

Author Response

The paper is important, will be more important if you also attach the code, and make it more general not just to prothesis but to any visual attention acuasition

Thanks for the comment.  I agree it might be suitable for a much wider class of circumstances that just prosthetics.  However as a reviewer myself, I am weary of overselling any development.  I also agree that the text needed up grading, thanks for the insight.

Still, it is not ready for publication - requires major majot revision

Agreed.

As a result there are too many changes to make tracked changes viable, hence it is an entirely separate script.

Abstract not very clear

Revised:

Gaze tracking, where the point of regard of a subject is mapped onto the image of the scene the subject sees, can be employed to study the visual attention of the users of prosthetic hands.  It can show whether the user is pays greater attention to the actions of their prosthetic hand as they use it to perform manipulation tasks, compared with the general population.  Conventional analysis of the video data requires a human operator to identify the key areas of interest in every frame of the video data.  Computer vision techniques can assist with this process, but a fully automatic systems requires large training sets.  Prosthetic investigations tend to be limited in numbers.  However, if the assessment task is well controlled, it is possible to make a much simpler system that uses initial input from an operator to identify the areas of interest and then the computer tracks the objects throughout the task.  The tool described here, employs colour separation and edge detection on images of the visual field to identify the objects to be tracked.  To simplify the computer's task further, this test uses the Southampton Hand Assessment Procedure (SHAP), to define the activity spatially and temporarily, reducing the search space for the computer.  The work reported here is the development a software tool capable of identifying and tracking the Points of Regard and Areas of Interest, throughout an activity with minimum human operator input.  Gaze was successfully tracked for fourteen unimpaired subjects, which was compared with the gaze of four users of myoelectric hands.  The SHAP cutting task is described and the differences in attention observed with a greater number of shorter fixations by the prosthesis users compared to unimpaired subjects.  There was less looking ahead to the next phase of the task by the prosthesis users

Define better the problem: what are the input of the problem?  Video of the face? The scene? The location of the hand? Iassume it is both from an eyetracker – so write it down.

Changed and added to at many points in the script.

Make the sections as two:

Introduction

Related work – if you want

But introduction need to be part of the introduction

The sections:  Background, Assessment Framework and Task are now subsections of the induction.  This is because the need to introduce very different aspects of the work.

Missing in introduction

Contribution of your work – not clear! You need to refer the contribution to existing similar competing teghcnioglies

Added although the point is that no one has chosen to adopt this process in eye tracking, so it has to come from the Kinect.

Methods

-          please add acheme of the experiment: human + eye tracker + scene

Figures 1 and 2 and associated text

-          Please add flow of the algorithm in a figure

Figures 4 and 5 and associated text

-          No formulas. No clear algorithm

Figure 6 and associated text

The majority of other decisions are at a very low definition of formula or algorithm and it could appear to the reader to over selling some straightforward ideas.  Again experience as a fellow reviewer makes me resistant to hyping up simple ideas.

Reviewer 2 Report

This paper presented an affordable system that is designed to speed up the process of processing eye-tracking data for the analysis of prosthetic hand function, using a standardized setup. A case study was conducted with fourteen normal subjects and four users of prosthetic limbs. My comments are:

1.     The authors should highlight the contributions in the Introduction.

2.     It is not easy to follow the experiment procedures. It is suggested to add an overall framework to show the whole processing steps.

3.     There are a few typos in the paper. For instance, line 98: “form the video stream”, line 230: “The the mean…”, line 241: “The the colour…”, line 333: “what is seen [?]” The authors need to thoroughly proofread the paper.

4.     It is very interesting to see the future work in the conclusion. The authors can provide a brief discussion. 

There are a few typos in the paper. For instance, line 98: “form the video stream”, line 230: “The the mean…”, line 241: “The the colour…”, line 333: “what is seen [?]” The authors need to thoroughly proofread the paper.

Author Response

This paper presented an affordable system that is designed to speed up the process of processing eye-tracking data for the analysis of prosthetic hand function, using a standardized setup. A case study was conducted with fourteen normal subjects and four users of prosthetic limbs.

Thanks for the comment.  I agree that the text needed up grading, thanks for the insights.  As a result there are too many changes to make tracked changes viable, hence it is an entirely separate script.

  1. The authors should highlight the contributions in the Introduction.

Added:

This paper describes a system designed speed up the process of processing eye tracking data for the analysis of prosthetic hand function. The program uses a standardised set up, including an activity of daily living tasks.  A computer is able to identify the key areas of interest and track them in the visual field throughout the activity, with minimal human intervention.  It is believed this is the first time this assisted analysis has been created for this application.

  1. It is not easy to follow the experiment procedures. It is suggested to add an overall framework to show the whole processing steps.

Indeed, apologies.  Text is rearranged and Figures 1 to 5 detail different aspects of the process from overall schematic to process flow.

  1. There are a few typos in the paper. For instance, line 98: “form the video stream”, line 230: “The the mean…”, line 241: “The the colour…”, line 333: “what is seen [?]” The authors need to thoroughly proofread the paper.

Apologies.  I was able to update the reference with a more recent one.

  1. It is very interesting to see the future work in the conclusion. The authors can provide a brief discussion.

Added to discussion rather than conclusion as some reviewers might regard the conclusion as being a place only for facts that have been supported by the results.  In addition the results of the use of the interface will be within a separate paper.

Refinements

While the system proved satisfactory there are a number of possible refinements to automate the system further:

 It is possible to detect trunk sway and measure its timing or extent from the data.  Again the choice of a SHAP task makes it far simpler as the displacement of the button from the lower centre of the visual field combined with the angle of the form board, is sufficient to detect the motion without needing to derive the precise angle of flexion. 

 The system does not rely on any particular form of visual information, such as how the PoR is defined or shown on the images which is a particular characteristic of each gaze tracking system.,  This tool has been shown to work with other gaze tracking hardware (Dikablis Professional Wireless Eye Tracker, (https://ergoneers.com/en/eye-tracker/dikablis-hdk-eye-tracker/). 

A refinement to the code would be to use the context of the known activity (such as cutting) to automatically search for the knife, button and plasticine in the early image based on their known colour where they should be in any SHAP sequence, although this level of automation will make the tool less general for use with other tasks.  Additionally, it is possible to use the motion data of prior frames to restrict the search box further, predicting position based on the preceding motion of the last few frames, rather than an arbitrary constant maximum speed.  Again this level of sophistication was not found to be needed in this application.

 This interface has been used to automate the process of studying gaze during the learning process.  When performing an unfamiliar task  subjects tend to fixate for shorter periods of time compared with routine operations, thus it is more important to have objective measures of gaze as provided by this program.

Round 2

Reviewer 1 Report

Much better

1. Fig 2 write instead of possible input additional input

2. figure 4,5  - make them as one - with two subplots - verify the fonts is fine

3. consider in the results adding histogram + more error statistics

Author Response

Much better

Thanks

  1. Fig 2 write instead of possible input additional input

Done

  1. figure 4,5 - make them as one - with two subplots - verify the fonts is fine

Done

  1. consider in the results adding histogram + more error statistics

Sorry, but I am not sure which histogram needs adding.  This paper is about the method and detailing more tasks would change the emphasis of the paper and make it harder to follow. 

We have added a measure of the range of glances to the graph (and at the same time corrected the y-axis legend), and with it slightly more information over the total number of glances that were measured in each group.  It certainly captures the differences between the groups more clearly.

We have been very careful not to conclude more than can be supported, we did not to over play the findings, that this interface works and makes the job easier.